# Autonomous State Estimation and Observability Analysis for the Taiji Formation Using High-Precision Optical Sensors

**DOI:** 10.3390/s23218672

**Published:** 2023-10-24

**Authors:** Bo Wen, Wenlin Tang, Xiaodong Peng, Zhen Yang

**Affiliations:** 1School of Fundamental Physics and Mathematical Sciences, Hangzhou Institute for Advanced Study, UCAS, Hangzhou 310024, China; wenbo21@mails.ucas.ac.cn (B.W.); pxd@nssc.ac.cn (X.P.); 2National Space Science Center, Chinese Academy of Sciences, Beijing 100190, China; yangzhen@nssc.ac.cn; 3University of Chinese Academy of Sciences, Beijing 100049, China; 4Taiji Laboratory for Gravitational Wave Universe, Hangzhou 310024, China

**Keywords:** observability analysis, autonomous state estimation, space-based gravitational wave detection formation, cubature Kalman filter

## Abstract

In certain observation periods of navigation missions for the Taiji formation, ground observation stations are unable to observe the spacecraft, while the state of the spacecraft can be estimated through the utilization of dynamic equations simulated on prior knowledge. However, this method cannot accurately track the spacecraft. In this paper, we focus on appropriately selecting the available onboard measurement to estimate the state of the spacecraft of the Taiji formation. We design two schemes to explore the performance of the state estimation based on the interspacecraft interferometry measurements and the measurements obtained from the Sun sensor and the radial velocity sensor. The observability of the system is numerically analyzed using the singular value decomposition method. Furthermore, we analyze error covariance propagation using the cubature Kalman filter. The results show that using high-precision interspacecraft angle measurement can improve significantly the observability of the system. The absolute position and velocity of the spacecraft can be estimated respectively with an accuracy of about 3.1 km and 0.14 m/s in the first scheme, where the prior information of the precision of the position and velocity is respectively 100 km and 1 m/s. When the measurement from the radial velocity sensor is used in the second scheme, the estimation accuracy of the velocity can be improved about 18 times better than that in the first scheme.

## 1. Introduction

The Taiji mission [1] is a type of space-based gravitational wave (GW) detection program that aims to detect GWs within a frequency range spanning from 0.1 mHz to 1.0 Hz. The Taiji mission consists of three spacecraft (SC), forming an SC formation known as the Taiji formation. Each SC follows a heliocentric orbit, forming a massive equilateral triangle with sides spanning approximately three million kilometers. Generally, navigation methods for such formations, which are located in heliocentric orbits, rely on a combination of deep space network (DSN) and other ground observations to estimate the absolute state of the SC [2,3]. However, due to the limitation of the observation arc, neither of these methods can provide continuous coverage for the entire observation period of the formation. As the Taiji formation is like that of LISA [4], we can borrow from the coverage analysis of observation of LISA. In the case of the LISA formation, there may be only a few hours a week of ground-based observation time, and the state of the SC is usually estimated based on the evolution of dynamical equations and a priori knowledge in the absence of ground observations [5]. These methods cannot track the SC well, especially when the SC performs unpredictable maneuvers. However, in the Taiji mission, each SC is equipped with a high-precision laser interspacecraft link measurement system [6] and other sensors; thus, these measurements may be used to perform the state estimation. These measurements include a laser measurement system to measure the relative ranges and range rates between SCs, and a differential wavefront sensing (DWS) system to measure the relative angles of SCs. Further, each SC is also equipped with a digital Sun sensor to determine the relative pointing angle between the SC and Sun. It is also possible to carry a spectrometer sensor to measure the radial velocity of the SC relative to Sun. Therefore, in this paper, we will focus on how to combine the available onboard measurements to autonomously estimate the state of the SC when the ground observations are insufficient.

The autonomous state estimation problem refers to estimating the SC state using only onboard measurement information. For the Taiji formation, there is currently no existing literature addressing the autonomous state estimation of the formation based on onboard sensors of the SC. Research approaches from another space-based GW detection formation with a design similar to the Taiji formation include the LISA formation [7], which can serve as valuable references. For both of these formations, the current approaches are to perform state estimation or orbit determination of the SC typically involving utilizing ground-based measurements (such as DSN and very long baseline interferometry (VLBI) measurement) in combination with interspacecraft measurements. For the LISA formation, Chung [2] investigated the absolute orbit determination and relative orbit determination of the SC utilizing ground observations and measurements of relative range and relative range rate between the SCs. This paper shows that when attempting to estimate the state of the SC solely based on interspacecraft range and range rate measurements, the filtering results would diverge, and the system is unobservable. Wang et al. [8] discovered that relying solely on interspacecraft measurements yields poor estimation of the absolute position state, while relative distance estimation remains accurate. For other formations with similar configurations, Hu et al. [9] found that the observability of the system using angles-only measurements is enhanced when there is a significant difference in eccentricity or inclination among SC orbits. On the other hand, studies [10,11,12] focusing on other interspacecraft measurements, such as relative range or relative range rate, indicate that using these measurements alone can determine the initial absolute orbit of the SC. However, when the SC orbits are coplanar, the state of the system using range or range rate measurement is usually unobservable.

Based on the aforementioned study on autonomous state estimation, we find that the observability of the system only using interspacecraft measurements becomes poor when the difference in eccentricity or inclination of SC orbits is small. For the Taiji or LISA formation, their nominal Keplerian orbit elements also exhibit these features, such as having the same inclination and eccentricity [13,14]. Consequently, the autonomous state estimation system may become unobservable or weakly observable in such cases. Studies [15,16,17] show that the addition of interspacecraft measurements can improve the accuracy of the state estimation. In addition to interspacecraft measurements, other onboard sensors can also be utilized. In these papers, they also show that, when using the measurements from the Sun sensor and the spectrometer sensor, the estimation accuracy of the absolute state of the SC can also be improved. Thus, like the study [18] that proposes a sensor selection algorithm to assess and compare the effects of different sensors on the accuracy of state estimation, for the Taiji mission, choosing the right sensors to model observation schemes for estimating the state of the SCs is an important study for the mission design.

In this paper, we investigate the problem of autonomous state estimation in the Taiji formation by utilizing all available onboard sensors. We specifically address state estimation in a multibody perturbation environment. Considering the payload configuration and sensor costs, for instance, the Taiji formation may opt not to equip a spectrometer for measuring relative velocity. We establish two observation schemes: the first case contains all measurements except the radial velocity of the spacecraft relative to Sun, and the second case contains all measurements. Furthermore, we explore the impact of high-precision optical angle sensors on the observability of the system using the singular value decomposition (SVD) method. Our findings demonstrate that the inclusion of high-precision angle sensors can enhance the observability of the system. Additionally, we analyze the propagation of error covariance of the state using the square cubature Kalman filter, particularly in a scenario with poor prior knowledge. Through our research, we achieve an estimated accuracy of approximately 3 km for the absolute position of the SC and approximately 0.1 m/s for the absolute velocity, where the prior information of the precision of the position and velocity is respectively 100 km and 1 m/s. These results contribute valuable a priori knowledge for the tasks in the Taiji mission, which need precision state information of SCs.

This paper is structured as follows: In Section 2, we introduce the dynamics and observation models of the Taiji mission. Section 3 focuses on the analysis of system observability using different schemes and measurement accuracies. In Section 4, we design a square root format of the cubature Kalman filter and then perform the simulation experiments of state estimation for the Taiji mission. Finally, Section 5 provides a summary of the main results of this paper.

## 2. System Model

The Taiji formation consists of three identical spacecraft (SC1, SC2, SC3). These three SCs form an approximately equilateral triangle formation with a side length of $3\times 10^{6}$ km, orbiting Sun [19], as shown in Figure 1. The reference coordinate frame for the formation is selected from the solar center of a mass inertial coordinate system. In this coordinate system, the origin is positioned at the center of the mass of Sun, the fundamental plane aligns with the plane of the ecliptic, the x-axis points towards the equinox, the y-axis lies within the fundamental plane and is perpendicular to the x-axis, and the z-axis extends perpendicular to the origin, following the right-hand rule.

The nonlinear dynamical model characterizing the Taiji formation is expressed as follows:(1)$$\left\{\begin{array}{cc} \dot{x}\left(t\right) & =f(x(t))+w(t)\\ y(k) & =h(x(k))+v(k) \end{array}\right..$$
where x represents the state vector of the system, f(x(t)) corresponds to the nonlinear continuous dynamical model of the system, w(t) represents the process noise of the system, h(x(k)) denotes the discrete observation model of the system, and v(k) represents the measurement noise of the state. A comprehensive explanation of these variables will be provided in the subsequent subsections.

### 2.1. Dynamics Model

The state vector of the system x is defined as follows:(2)$$\begin{array}{cc} r_{i} & ={\left[x_{i},y_{i},z_{i}\right]}^{T}\;\;\left(i=1,2,3\right), \end{array}$$(3)$$\begin{array}{cc} \dot{r_{i}} & ={\left[\dot{x_{i}},\dot{y_{i}},\dot{z_{i}}\right]}^{T}\;\;\left(i=1,2,3\right), \end{array}$$(4)$$\begin{array}{cc} x & ={\left[r_{1},r_{2},r_{3},\dot{r_{1}},\dot{r_{2}},\dot{r_{3}}\right]}^{T}\in R^{18}, \end{array}$$
where *i* represents the index of the SC in the formation, $r_{i}$ represents the position state of the SC in Cartesian form, and $\dot{r_{i}}$ represents the velocity state of the SC in the coordinate frame.

f(x(t)) represents the nonlinear dynamical equations of the system:(5)$$\begin{array}{cc} f(x(t)) & =\left[\begin{array}{c} \dot{r_{1}}\\ \dot{r_{2}}\\ \dot{r_{3}}\\ \ddot{r_{1}}\\ \ddot{r_{2}}\\ \ddot{r_{3}} \end{array}\right]\in R^{18}. \end{array}$$
where
(6)$$\ddot{r_{i}}=-\frac{\mu_{s}}{r_{i}^{3}}r_{i}+\sum_{j=1}^{n_{p}}\mu_{j}\left[\frac{r_{pj}-r_{i}}{∥r_{pj}-r_{i}{∥}^{3}}-\frac{r_{pj}}{r_{pj}^{3}}\right]\;\;\left(i=1,2,3,j=1,2,3,\ldots ,6\right),$$
where $\mu_{s}$ represents the gravitational constant of Sun, $r_{i}=\sqrt{x_{i}^{2}+y_{i}^{2}+z_{i}^{2}}$ represents the range from the *i*-th SC to the center of Sun, $\mu_{j}$ represents the gravitational constant of the *j*-th perturbing planet, $r_{pj}$ represents the heliocentric coordinate system position vector of the *j*-th perturbing planet, $r_{pj}-r_{i}$ represents the position vector of the *j*-th perturbing planet relative to the *i*-th SC, and $n_{p}=6$ is the number of perturbing planets. For each SC, we have considered the gravitational influence of six celestial bodies, including Mercury, Venus, Earth, Mars, Jupiter, and Saturn.

w(t) and v(t) are modeled as the zero-mean Gaussian white noise and are not related to each other:(7)$$\left\{\begin{array}{cc} w(t) & ∼(0,Q_{c})\\ v(k) & ∼(0,R_{k})\\ E(w\left(t\right)w{\left(t-\tau \right)}^{T}) & =Q_{c}\delta \left(t-\tau \right)\\ E(v\left(k\right)v{\left(k-\tau \right)}^{T}) & =R_{k}\delta \left(k-\tau \right)\\ E(w\left(t\right)v{\left(k\right)}^{T}) & =0 \end{array}\right..$$
where $Q_{c}$ is the covariance of the process noise distribution, and $R_{k}$ is the covariance of the measurement noise distribution.

### 2.2. Observational Model

In our study, we employ a high-precision interspacecraft measurement system along with other onboard sensors on the SC to gather measurement information for the purpose of state estimation. The interspacecraft measurements for the SC primarily encompass relative range measurements, relative range rate measurements, and relative angle measurements. In addition, we make use of the digital Sun sensor and spectrometer installed on the SC as supplementary measurement sensors. It is important to note that the measurement information considered in our study undergoes preprocessing and time synchronization. Consequently, we provide detailed descriptions of the specific instrumentation used and present simplified mathematical models for interpreting these measurement data.

#### 2.2.1. Interspacecraft Measurements

Space-based GW detection formations employ a laser interferometry instrument [20] with extremely high measurement accuracy to capture the information between spacecraft. For the sake of convenience, we initially denote the relative position and relative velocity vectors between the spacecraft:(8)$$\begin{array}{cc} r_{\mathrm{ij}} & =[x_{i}-x_{j},y_{i}-y_{j},z_{i}-z_{j}], \end{array}$$(9)$$\begin{array}{cc} \dot{r_{\mathrm{ij}}} & =[\dot{x_{i}}-\dot{x_{j}},\dot{y_{i}}-\dot{y_{j}},\dot{z_{i}}-\dot{z_{j}}], \end{array}$$
where *i* and *j* represent the indices of the SC. The relative range between the *i*-th SC and the *j*-th SC is measured by the laser ranging system, which can be modeled as
(10)$$\begin{array}{c} L_{ij}=\left|\right|r_{\mathrm{ij}}\left|\right|, \end{array}$$

The relative range rate measured by the Doppler shift measurement system can be modeled as
(11)$$\dot{L_{ij}}={\left(\frac{r_{\mathrm{ij}}}{\left|\right|r_{\mathrm{ij}}\left|\right|}\right)}^{T}\dot{r_{\mathrm{ij}}}.$$

The relative angles between spacecraft are measured using the DWS system, a widely recognized technique for precisely quantifying the relative wavefront misalignment between two beams with high sensitivity. In this context, the measured relative angles include the azimuth angle denoted by λ and the elevation angle denoted by ϕ, which are generally measured using CCD and quadrant photodiode (QPD) sensors. In the Taiji program, relative angles can be measured with a very high accuracy of 1 nrad [21,22]. The DWS measurement of these angles can be modeled as follows:(12)$$\begin{array}{cc} \lambda_{ij} & =arctan(\frac{y_{ij}}{x_{ij}}), \end{array}$$(13)$$\begin{array}{cc} \phi_{ij} & =arcsin(\frac{z_{ij}}{\left|\right|r_{\mathrm{ij}}\left|\right|}), \end{array}$$
where $x_{ij}=x_{i}-x_{j}$ and $y_{ij}$, $z_{ij}$ are similar to it. Moreover, the relative angle measurements can also be alternatively modeled using the light-of-sight (LOS) vector, which provides an equivalent representation:(14)$$A_{ij}=\frac{r_{\mathrm{ij}}}{\left|\right|r_{\mathrm{ij}}\left|\right|}.$$

#### 2.2.2. Other Measurements Onboard the Spacecraft

The measurement of the relative Sun-pointing angles between Sun and the *i*-th SC is obtained using the digital Sun sensor, which can be modeled as follows:(15)$$A_{si}=\frac{r_{i}}{\left|\right|r_{i}\left|\right|},$$

The radial velocity can be measured using the spectrometer on the SC, which detects the Doppler shift caused by the relative motion between Sun and the *i*-th SC. This measurement can be modeled as follows:(16)$$V_{i}={\left(\frac{r_{i}}{\left|\right|r_{i}\left|\right|}\right)}^{T}\dot{r_{i}}.$$

#### 2.2.3. Measurement Equations

Considering the payload constraints of the SC, we have developed two observation model schemes as demonstrated in Table 1. The first scheme, referred to as case 1, does not include the radial velocity information provided by the spectrometer. On the other hand, the second scheme, known as case 2, incorporates this additional measurement.

These two observation models of the system can be written as
(17)$$h_{1}\left(x\right)={\left[L_{12},L_{13},L_{23},\dot{L_{12}},\dot{L_{13}},\dot{L_{23}},A_{12},A_{13},A_{23},A_{s1},A_{s2},A_{s3}\right]}^{T}\in R^{24},$$
(18)$$h_{2}\left(x\right)={\left[h_{1}\left(x\right),V_{1},V_{2},V_{3}\right]}^{T}\in R^{27}.$$

## 3. Observability Analysis

The observability analysis of a system provides a crucial theoretical foundation for state estimation. It helps determine whether the system’s state can be accurately estimated based on available measurement information. If the system is observable, the convergence of the state error occurs gradually as the measurement information is continually updated. The observability of nonlinear systems is typically assessed using the rank criterion approach, where the rank of the observability matrix serves as an indicator of the system’s observability level. In this study, we employ the observability Gramian matrix [23] to compute the system’s observability matrix. Specifically, for discrete systems, the observability Gramian matrix without considering measurement noise can be expressed as follows:(19)$$\begin{array}{c} G_{0,n}=\sum_{k=0}^{n}\Phi_{0,k}^{T}H_{k}^{T}H_{k}\Phi_{0,k}, \end{array}$$
where *n* represents the total number of measurement instances starting from $t_{0}$. $\Phi_{0,k}$ denotes the system transfer matrix from $t_{0}$ to $t_{k}$, while $H_{k}$ represents the Jacobian derivative matrix derived from the observation model at time $t_{k}$.
(20)$$\begin{array}{cc} A_{k} & =\frac{\partial f}{\partial x}{|}_{x_{k}}, \end{array}$$
(21)$$\begin{array}{cc} H_{k} & =\frac{\partial h}{\partial x}{|}_{x_{k}}, \end{array}$$
(22)$$\begin{array}{cc} \Phi_{k} & =e^{A_{k}\Delta t}, \end{array}$$

In the provided equation, for small sampling times Δt, the system transfer matrix $\Phi_{k}$ can be approximated as $\Phi_{k}=I+\Delta tA_{k}$. To numerically quantify the initial observability of the system at time $t_{k}$, we employ the singular value decomposition (SVD) method to analyze the observability Gramian matrix $G_{0,n}$ as follows:(23)$$\begin{array}{c} G_{0,n}=U\Sigma V^{T}. \end{array}$$

In the last equation, the matrix Σ represents the diagonal matrix composed of the singular values of the observability Gramian matrix $G_{0,n}$, while *U* and *V* correspond to the left and right unitary matrices, respectively. We employ two metrics to measure the degree of observability of the system:(24)$$\begin{array}{cc} OI & =\sigma_{min}, \end{array}$$(25)$$\begin{array}{cc} CN & =\frac{\sigma_{max}}{\sigma_{min}}. \end{array}$$

The variable $\sigma_{min}$ represents the smallest individual singular value obtained from the matrix Σ, and $\sigma_{max}$ represents the largest individual singular value obtained from that. The observability index (OI) signifies the measure of observability for the observability matrix, while the condition number (CN) denotes the numerical condition of the observability matrix. A lower OI value or a higher CN value signifies reduced system observability. These two indicators serve as evaluative metrics to assess the degree of observability of the system. For our study, we numerically calculate these two metrics, OI and CN, for both case 1 and case 2, using the initial state of the system. The initial state vector for each SC within the Taiji formation in the inertial frame can be derived from the initial Keplerian orbital elements [24].

One initial Keplerian orbital parameter for the Taiji formation is present in Table 2, encompassing the semimajor axis *a*, eccentricity *e*, inclination *i*, right ascension of the ascending node Ω, argument of the perihelion ω, and true anomaly *f*. To determine the true value of the measurement information, we calculate it indirectly based on the state value at the initial time $t_{0}$. For both case 1 and case 2, we compute the condition number (CN) and observability index (OI) of the initial observability Gramian matrix $G_{0,0}$ as demonstrated in Table 3.

Our findings indicate that the observability of the system employing an observation scheme similar to case 1 is slightly weaker compared with the system employing the case 2 scheme. This disparity may be attributed to the principle that the inclusion of a greater variety of observation types enhances the effectiveness of state estimation.

The observability of a system is also influenced by the accuracy of the measurement information. For instance, Geller et al. [25] discovered that when the angle measurement accuracy is on the order of 1 mrad, certain initial orbit determination challenges become difficult to resolve. Additionally, Yim et al. [26] found that there is a linear relationship between the observability of a system utilizing angles-only measurements and the angle measurement accuracy. In the Taiji formation, each SC is equipped with high-precision angle measurement sensors, such as CCD and QPD. In this study, our focus lies in investigating the extent to which the accuracy of the angle measurement sensor impacts the observability of the system.

We also use the observability Gramian matrix with measurement noise to investigate this issue:(26)$$\begin{array}{c} L_{0,n}=\sum_{k=0}^{n}\Phi_{0,k}^{T}H_{k}^{T}R_{k}^{-1}H_{k}\Phi_{0,k} \end{array}$$

Here, $R_{k}$ represents the measurement covariance of the system at time *k*. To quantify the impact of measurement accuracy on the observability of the system, we employ a ratio metric proposed by Hu et al. [9] to measure the effect of measurement accuracy on the observability of the system, where a smaller value indicates weaker observability of the system.
(27)$$OI_{r}=\frac{OI_{L}}{OI_{G}}=\frac{min\;\sigma (L_{0,k})}{min\;\sigma (G_{0,k})}.$$

In our research, we focus on the case 2 observation scheme, which involves the utilization of almost all available measurement sensors on the SC. To assess the observability of the system, we compare three levels of angle measurement accuracy. These accuracy levels represent different types of sensors: low-precision sensors, reflecting current sensor capabilities; common sensors, planned to be installed on the SC; and high-precision sensors that may be used in the future. These accuracy levels are separated by a factor of $10^{3}$. We assume that the measurement noise of the sensors follows a Gaussian distribution, and the standard deviation of the sensors for each level is presented in Table 4:

To assess and compare the observability of the system, we calculate the observability index ratio ($OI_{r}$) using 1000 data points within each interval. The measurement standard deviation step for each level is set as $1\times 10^{-9}$ from level 3 to level 2. From level 2 to level 1, the measurement standard deviation step for a sensor is set as $1\times 10^{-6}$. The $OI_{r}$ values for the system with varying measurement accuracy are presented below:

Table 5 below presents OI and CN of the observability matrix of the system at different levels of measurement accuracy.

From Figure 2, we observe that the degree of observability of the system demonstrates a nearly linear growth pattern as the accuracy of angle measurement increases, particularly from level 2 to level 1. However, we find that the observability of the system shows minimal improvement from level 3 to level 2, despite the increase in angle measurement accuracy from approximately $3\times 10^{-9}$ rad to $1\times 10^{-9}$ rad. There are several factors contributing to these observations. First, in the absence of ground-based observations, the system tends to exhibit weak observability. Additionally, the unique orbit configurations of the Taiji formation, such as having orbits with similar inclinations or eccentricities, also contribute to the reduced observability of the system.

Through the above analysis, we find that high-precision angle measurement information can significantly improve the observability of the system. In the Taiji formation, each SC is equipped with high-precision angle sensors like CCD and QPD. To further investigate the impact of angle measurement accuracy on system observability, numerical simulation experiments were conducted using the nonlinear Kalman filter as described in Section 4. Additionally, a comparative evaluation of the state estimation accuracy was performed for two observation schemes.

## 4. Numerical Simulation

### 4.1. Simulation Initialization

Since the Taiji formation has not yet been launched, there is a lack of real data available for research. We employ a numerical simulation method to investigate the problem of autonomous state estimation. The initial state of the formation is computed using the Keplerian orbits, as illustrated in Table 2. Subsequently, we employ the fourth-order Runge–Kutta method to solve differential Equation (Equation 5) and simulate the true state of the SC from the initial time. For this simulation, we use a discrete step length of 1/3 s. The true measurement values for case 1 and case 2 at each time instant *k* are computed using Equations (Equation 17) and (Equation 18), respectively. It is assumed that the measurement information has been preprocessed, including time synchronization. In the state estimation process, the measurement information employed comprises the true measurement value with added noise. The process noise w(t) and the measurement noise v(k) are assumed to adhere to Equation (Equation 7). Specific noise deviations and simulation parameters are provided in Table 6:

The covariance matrix of the process noise, denoted as *Q*, is set as $Q=\mathrm{diag}(\left[I_{1\times 18}\times 10^{-12}\right])$. Similarly, the measurement noise matrix, denoted as *R*, is set as $R=\mathrm{diag}(\left[I_{1\times 3}\cdot 1,I_{1\times 3}\cdot 10^{-12},I_{1\times 9}\cdot 10^{-12},I_{1\times 9}\cdot 10^{-10},I_{1\times 3}\cdot 10^{-4}\right])$. In general, the a priori estimation error of the spacecraft’s position state is assumed to be 10 km, while the a priori knowledge error of the velocity state is 0.1 m/s. However, to consider a worst-case scenario with less accurate prior state knowledge, we increase the uncertainty by a factor of 10. Consequently, the covariance matrix of the state uncertainty is set as $P_{0}=\mathrm{diag}\left(\left[I_{1\times 9}\cdot 10^{10},I_{1\times 9}\cdot 1\right]\right)$.

We employ the cubature Kalman filter (CKF) [27,28] to estimate the state, leveraging its robustness against divergence in highly nonlinear systems. Considering numerical stability, we adopt the square root form of the filter to perform covariance analysis in the case of weakly observable systems. The algorithmic process of the square root cubature Kalman filter (SR-CKF) for a hybrid system, consisting of continuous dynamics equations and discrete observation equations, can be described as follows:

Given the initial state estimation ${\hat{x}}_{0|0}$ and the covariance matrix of the uncertainty estimation $P_{0|0}$ obtained from prior knowledge, the initial square root of the error covariance $S_{0|0}$ can be computed using the Cholesky decomposition of $P_{0|0}$.
(28)$$\begin{array}{c} P_{0|0}=S_{0|0}S_{0|0}^{T} \end{array}$$

The time update process can be described as follows:1.Calculate the cubature points $X_{i,k-1|k-1}$ (where i=1,2,…,m) using the square root factor $S_{k-1|k-1}$ and the common cubature points $\zeta_{i}$:
(29)$$X_{i,k-1|k-1}=S_{k-1|k-1}\zeta_{i}+{\hat{x}}_{k-1|k-1}$$Here, *m* is twice the dimension $n_{x}$ of the state variable x, and $\zeta_{i}$ can be defined as
(30)$$\zeta_{i}=\left\{\begin{array}{ccc} \sqrt{\frac{m}{2}}\cdot \epsilon_{i} & \; & i=1,2\cdots n_{x}\\ -\sqrt{\frac{m}{2}}\cdot \epsilon_{i-n_{x}} & \; & i=n_{x}+1,n_{x}+2\cdots m \end{array}\right.$$
where $\epsilon_{i}$ denotes the i-th column of the unit matrix $I_{n_{x}}$.2.Evaluate the propagated cubature points:
(31)$$X_{i,k|k-1}^{*}=f\left(X_{i,k-1|k-1}\right)$$3.Estimate the predicted state:
(32)$${\hat{x}}_{k|k-1}=\frac{1}{m}\sum_{i=1}^{m}X_{i,k|k-1}^{*}$$4.Estimate the square root factor of the predicted error covariance:
(33)$$\begin{array}{cc} S_{k|k-1} & =\mathrm{qr}(\left[X_{k|k-1}^{*}\;S_{Q,k-1}\right]) \end{array}$$
(34)$$\begin{array}{cc} Q_{k-1} & =S_{Q,k-1}S_{Q,k-1}^{T} \end{array}$$Here, qr denotes the QR decomposition of the matrix, $S_{Q,k-1}$ represents the square root factor of the process noise matrix $Q_{k-1}$ at time *k*, and $X_{k|k-1}^{}$ is the centered weighted matrix, which can be calculated as
(35)$$\begin{array}{c} X_{k|k-1}^{*}=\frac{1}{\sqrt{m}}\left[X_{1,k|k-1}^{*}-{\hat{x}}_{k|k-1}\;X_{2,k|k-1}^{*}-{\hat{x}}_{k|k-1}\cdots X_{m,k|k-1}^{*}-{\hat{x}}_{k|k-1}\right] \end{array}$$

The measurement update process is as follows:1.Calculate the cubature points again:
(36)$$X_{i,k|k-1}=S_{k|k-1}\zeta_{i}+{\hat{x}}_{k|k-1}$$2.Evaluate the propagated cubature points:
(37)$$Y_{i,k|k-1}=h\left(X_{i,k|k-1}\right)$$3.Estimate the predicted measurement:
(38)$${\hat{y}}_{k|k-1}=\frac{1}{m}\sum_{i=1}^{m}Y_{i,k|k-1}^{*}$$4.Estimate the square root factor of the innovation covariance matrix:
(39)$$\begin{array}{cc} S_{yy,k|k-1} & =\mathrm{qr}(\left[Y_{k|k-1}\;S_{R,k-1}\right]) \end{array}$$
(40)$$\begin{array}{cc} R_{k-1} & =S_{R,k-1}S_{R,k-1}^{T} \end{array}$$Here, $S_{R,k-1}$ is a square root factor of the measurement noise matrix $R_{k-1}$ at time *k*, and $Y_{k|k-1}$ is the centered weighted matrix, calculated as
(41)$$\begin{array}{c} Y_{k|k-1}=\frac{1}{\sqrt{m}}\left[Y_{1,k|k-1}-{\hat{y}}_{k|k-1}\;Y_{2,k|k-1}-{\hat{y}}_{k|k-1}\cdots Y_{m,k|k-1}-{\hat{y}}_{k|k-1}\right] \end{array}$$Additionally, the centered weighted matrix $X_{k|k-1}$ can be calculated as
(42)$$\begin{array}{c} X_{k|k-1}=\frac{1}{\sqrt{m}}\left[X_{1,k|k-1}-{\hat{x}}_{k|k-1}\;X_{2,k|k-1}-{\hat{x}}_{k|k-1}\cdots X_{m,k|k-1}-{\hat{x}}_{k|k-1}\right] \end{array}$$5.Calculate the cross-covariance matrix:
(43)$$P_{xy,k|k-1}=X_{k|k-1}Y_{k|k-1}^{T}$$6.Estimate the Kalman gain:
(44)$$K_{k}=\left(P_{xy,k|k-1}/S_{yy,k|k-1}\right)/S_{yy,k|k-1}$$7.Update the state estimation:
(45)$${\hat{x}}_{k|k}={\hat{x}}_{k|k-1}+K_{k}\left(y_{k}-{\hat{y}}_{k|k-1}\right)$$Then the square root factor of the corresponding error covariance can be estimated as
(46)$$S_{k|k}=\mathrm{qr}\left(\left[X_{k|k-1}-K_{k}Y_{k|k-1}\;K_{k}S_{R,k}\right]\right)$$This state estimation algorithm is executed recursively for each sampling time, denoted as *k*. The simulation results are presented in the subsequent subsection, as demonstrated below.

### 4.2. Simulation Results

As depicted in Table 4, three levels of accuracy are considered for relative angle measurement. Initially, we conduct state estimation experiments on case 2, varying the accuracy of angle measurements, to validate the results obtained from the observability analysis of the system presented in Section 3. Subsequently, we compare the influence of case 1 and case 2 on the accuracy of state estimation to investigate the significance of the radial velocity sensor.

As illustrated in Figure 3, Figure 4 and Figure 5 and Table 7, we can see that the worst-case estimation errors for position and velocity states are approximately 2.0 km and 0.0078 m/s, respectively; under the highest relative angular measurement accuracy of 1 nrad, the state estimation outcomes for the three SCs in the Taiji formation exhibit similar trends. That may be due to the fact that the three spacecraft have similar orbital configurations and the same sensor configuration. We find that the estimation error of the position state of the SC gradually increases at level 1 but converges progressively at level 2 and level 3. The velocity state error remains stable across all three levels of measurement accuracy. This stability can be attributed to the direct availability of velocity measurement information from the radial velocity sensor, whereas the position state cannot be directly measured. Notably, as the relative angular measurement accuracy improves, the accuracy of SC state estimation also increases.

Considering the SC payload limitations and other practical considerations, it is possible that the SC in the Taiji formation may not be equipped with an extensive array of sensors specifically designed for solar measurement purposes, such as a spectrometer capable of measuring the spacecraft’s velocity with respect to Sun. To assess the impact of the radial velocity sensor on state estimation, we conduct identical simulation experiments in case 1. This allows for a direct comparison and evaluation of the effect of the radial velocity sensor on the accuracy of state estimation.

As illustrated in Figure 5 and Figure 6 and Table 7 and Table 8, we can see that the worst-case estimation errors for position and velocity states are approximately 3.1 km and 0.14 m/s, respectively, under the highest relative angular measurement accuracy of 1 nrad. When comparing the state estimation results between case 1 and case 2 with the same relative angle measurement accuracy, there is little difference in the accuracy of position state estimation of the SC. However, there is a significant disparity in the estimation of velocity of the SC. The velocity estimation results in case 2 are approximately 18 times more precise than those in case 1. This is likely due to the inclusion of radial velocity measurements in case 2, which directly provides velocity information of the SC, resulting in significant improvement of velocity estimation. Further, the filter is more stable in case 2 compared with that in case 1.

## 5. Conclusions

This paper presents an analysis of the autonomous state estimation problem of the Taiji formation. To assess the importance of a radial velocity sensor and considering the sensor configuration of the formation, we propose two observation schemes: case 1 contains all measurements except the radial velocity measurement, and case 2 contains all measurements. Considering that there are high-precision optical relative angle sensors in the Taiji formation, we study the influence of three distinct angular measurement accuracies on state estimation precision. By the observability analysis, we find that the utilization of high-precision angle measurement sensors can significantly enhance the observability of the system. By numerical simulation experiments, we find that, in case 2 with a relative angle measurement accuracy of 1 nrad, even with poor prior state knowledge, in which the prior information of the precision of the position and velocity is respectively 100 km and 1 m/s, the position of the SC can be estimated with an accuracy of approximately 2.0 km, while the velocity can reach approximately 0.0078 m/s. In the absence of radial velocity measurement in case 1, with a relative angle measurement accuracy of 1 nrad, the estimated accuracy of the position is approximately 3.1 km. However, the estimation accuracy of velocity is approximately 0.14 m/s. This could be attributed to the absence of measured velocity state information of the SC. It is noteworthy that the Taiji formation may not carry the spectrometer to measure the radial velocity of the SC relative to Sun. For certain missions, such as the alignment of laser between two distant SCs, of the Taiji formation, the sensor configuration of case 1 may also meet the requirements for SC state estimation accuracy, especially when the SC is equipped with a high-precision relative angle measurement sensor. The study proposed in this paper can be applied to address the autonomous state estimation problem when ground measurement information is unavailable and can assist in selecting the appropriate sensor configuration for the state estimation of the Taiji formation.

## Figures and Tables

**Figure 1 sensors-23-08672-f001:**
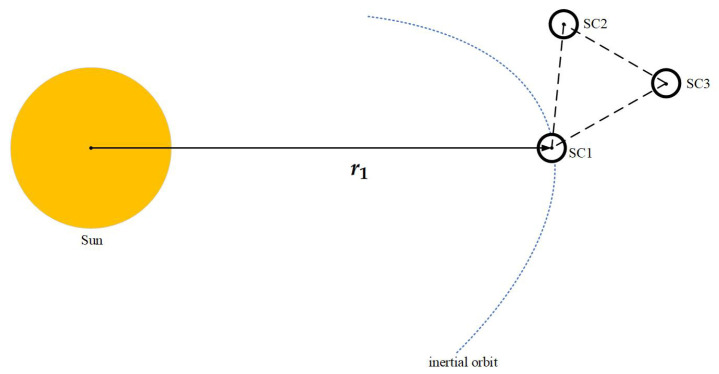
The Taiji formation.

**Figure 2 sensors-23-08672-f002:**
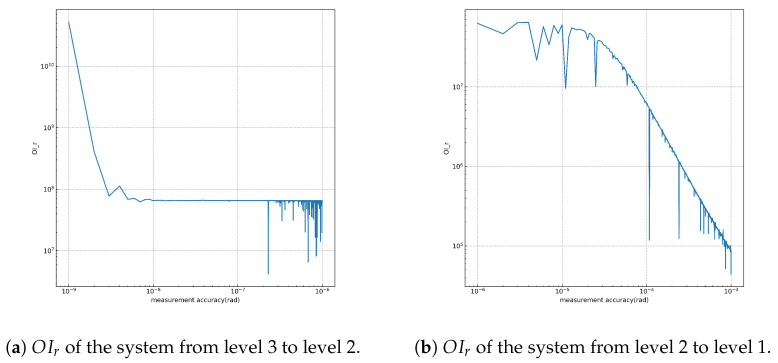
Observability measures with different angle measurement accuracy.

**Figure 3 sensors-23-08672-f003:**
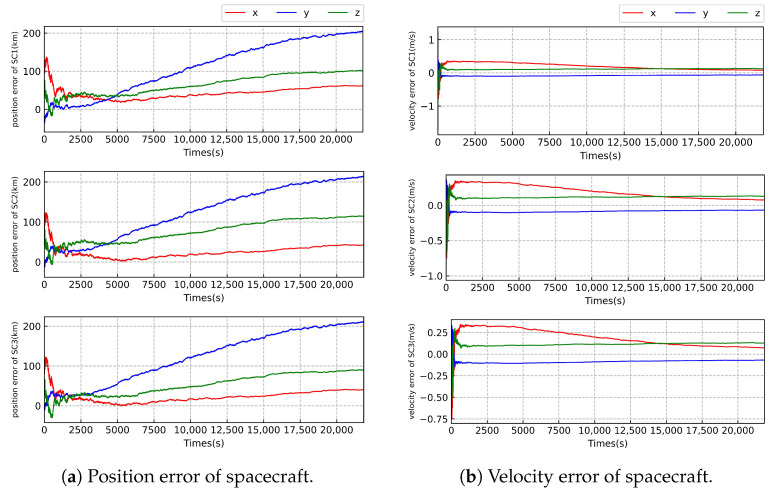
Error of state estimation of the Taiji formation with the measurement accuracy of level 1.

**Figure 4 sensors-23-08672-f004:**
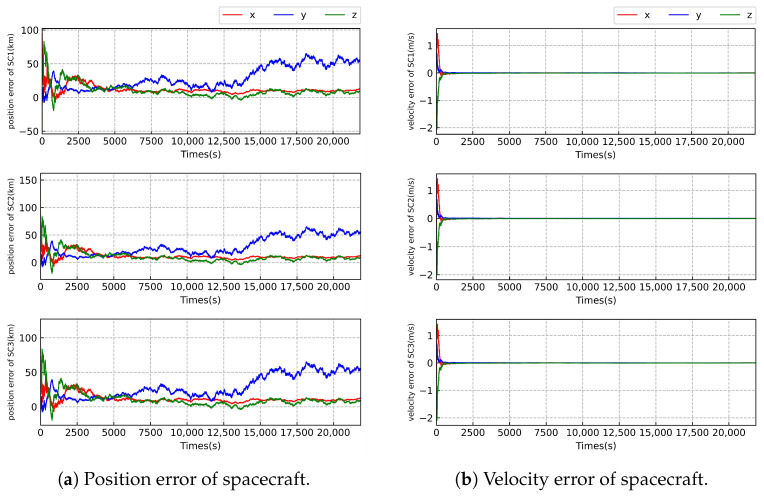
Error of state estimation of the Taiji formation with the measurement accuracy of level 2.

**Figure 5 sensors-23-08672-f005:**
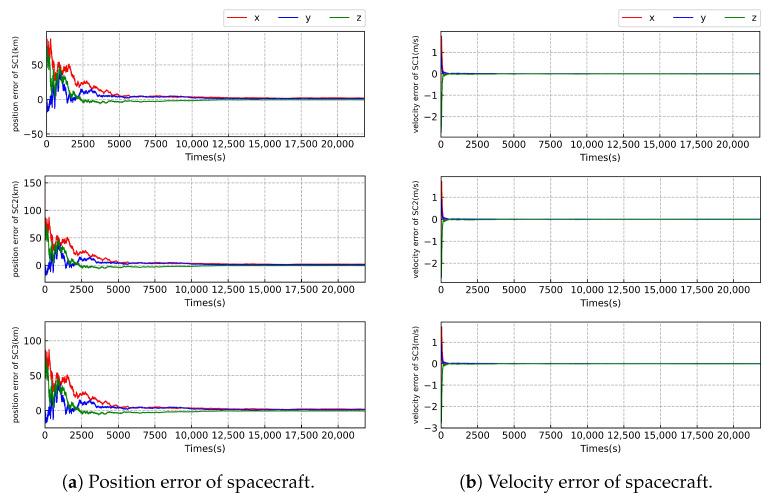
Error of state estimation of the Taiji formation with the measurement accuracy of level 3.

**Figure 6 sensors-23-08672-f006:**
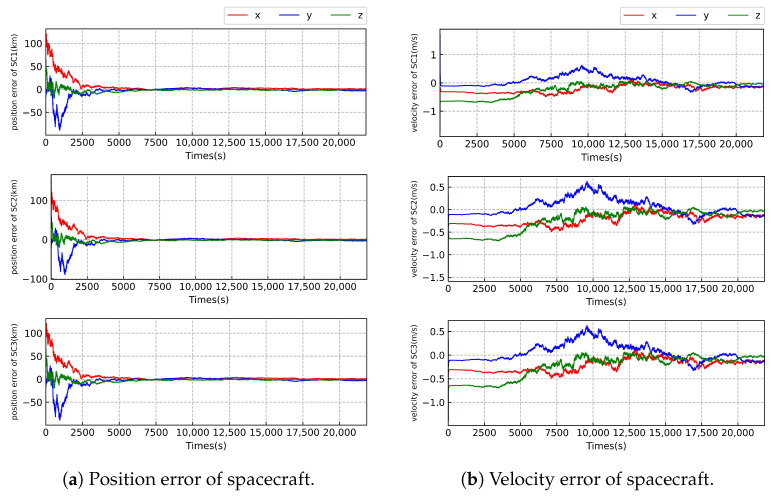
Error of state estimation of the Taiji formation with the measurement accuracy of level 3 in case 1.

**Table 1 sensors-23-08672-t001:** Measurement cases.

	Interspacecraft Measurements			Sun Angle Measurements	Radial Velocity Measurements
	**Range**	**Range Rate**	**Angle**		
case 1	✓	✓	✓	✓	
case 2	✓	✓	✓	✓	✓

**Table 2 sensors-23-08672-t002:** Initial Keplerian orbital elements for the Taiji formation.

	a (m)	e	i (rad)	Ω (rad)	ω (rad)	f (rad)
SC1	149596345656	0.0057765	0.41885699 04957726	6.2789100 17676982	4.5455776 28265889	3.14159265 3580112
SC2	149596558992	0.0057787	0.40563848 15646019	0.0236941 87073916	0.3314021 192709651	1.05695599 21295202
SC3	149596971939	0.0057800	0.40266269 23819615	6.2634647 12324839	2.4647099 56430478	−1.0565641 717284677

**Table 3 sensors-23-08672-t003:** The observability metrics for two observation schemes.

	OI	CN
case 1	$3.251\times 10^{-32}$	$9.227\times 10^{31}$
case 2	$2.404\times 10^{-31}$	$1.248\times 10^{31}$

**Table 4 sensors-23-08672-t004:** The measurement standard deviation of measurement sensors.

	Interspacecraft Measurements			Sun Angle Measurements	Radial Velocity Measurements
	**Range (m)**	**Range Rate (m/s)**	**Angle (rad)**	**Sun-Pointing Angle (rad)**	**Range Rate to Sun (m/s)**
level 1 (low-precision)	1	$1\times 10^{-6}$	$1\times 10^{-3}$	$1\times 10^{-5}$	$1\times 10^{-2}$
level 2 (common)			$1\times 10^{-6}$		
level 3 (high-precision)			$1\times 10^{-9}$		

**Table 5 sensors-23-08672-t005:** OI and CN for different measurement accuracy levels.

	OI	CN
level 1	$2.090\times 10^{-26}$	$1.436\times 10^{26}$
level 2	$1.542\times 10^{-23}$	$1.945\times 10^{23}$
level 3	$1.244\times 10^{-20}$	$2.412\times 10^{20}$

**Table 6 sensors-23-08672-t006:** The simulation parameters.

Parameter	Value
process noise standard deviation	$1\times 10^{-12}$
relative range noise standard deviation	1 m
relative range rate noise standard deviation	$1\times 10^{-6}$ m/s
relative angle noise standard deviation	$1\times 10^{-6}$ rad
Sun-pointing angle noise standard deviation	$1\times 10^{-5}$ rad
radial velocity noise standard deviation	$1\times 10^{-2}$ m/s
simulation time step	1/3 s
simulation duration	$2^{16}/3$ s
measurement frequency	once per 1/3 s

**Table 7 sensors-23-08672-t007:** The state estimation errors of the spacecraft in case 2.

	Measurement Level	*x* (km)	*y* (km)	*z* (km)	$\dot{x}$ (m/s)	$\dot{y}$ (m/s)	$\dot{z}$ (m/s)
SC1	level 1	61.089	202.740	100.296	0.077	−0.065	0.124
	level 2	12.657	55.100	9.132	$4.197\times 10^{-3}$	$0.987\times 10^{-3}$	$-2.164\times 10^{-3}$
	level 3	1.987	1.415	−0.746	$1.733\times 10^{-3}$	$3.475\times 10^{-3}$	$-7.778\times 10^{-3}$
SC2	level 1	41.785	212.048	113.512	0.076	−0.066	0.126
	level 2	12.650	55.104	9.136	$4.248\times 10^{-3}$	$1.174\times 10^{-3}$	$-2.203\times 10^{-3}$
	level 3	1.988	1.416	−0.747	$1.732\times 10^{-3}$	$3.480\times 10^{-3}$	$-7.767\times 10^{-3}$
SC3	level 1	39.591	209.159	88.955	0.073	−0.068	0.126
	level 2	12.647	55.088	9.130	$4.200\times 10^{-3}$	$0.818\times 10^{-3}$	$-2.131\times 10^{-3}$
	level 3	1.985	1.414	−0.745	$1.734\times 10^{-3}$	$3.478\times 10^{-3}$	$-7.768\times 10^{-3}$

**Table 8 sensors-23-08672-t008:** The state estimation errors of the spacecraft with the accuracy of level 3 in case 1.

	*x* (km)	*y* (km)	*z* (km)	$\dot{x}$ (m/s)	$\dot{y}$ (m/s)	$\dot{z}$ (m/s)
SC1	1.290	−3.120	−0.885	−0.103	−0.143	−0.024
SC2	1.291	−3.118	−0.883	−0.101	−0.142	−0.027
SC3	1.289	−3.121	−0.883	−0.103	−0.146	−0.022

## Data Availability

Data underlying the results presented in this paper are not publicly available at this time but may be obtained from the authors upon reasonable request.

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
