# Peer review of "Autonomous State Estimation and Observability Analysis for the Taiji Formation Using High-Precision Optical Sensors"

_sensors, 2023, doi:10.3390/s23218672_

Round 1
Reviewer 1 Report
The authors present autonomous state estimation method for the Taiji formation using the available onboard measurements when there is no ground measurements from the DSN or the VLBI network. They present two schemes of the combinations of onboard measurements to study the state estimation with different levels of measurement accuracies. They compare the performance of the two schemes and then suggest when we want to estimate the velocity more well, it is important to introduce the measurement of the radial velocity measurements from the spectrometer. The problem proposed is important for the design of the Taiji mission. It provides a useful solution to the autonomous state estimation of the spacecrafts of Taiji.
The study performed is motivated and rigorous in its implementation. The proposed publication journal is appropriate. However, this reviewer has a few 'Major issues' described below need to be corrected to consider this work for publication.
Major issues:
The language of the draft needs to be improved considerably. Although the language level is good enough to understand the work, it is not appropriate for publication in a journal like Sensors.
Lines 1-14: The aim of this paper is to study the autonomous state estimation of Taiji mission using the available onboard measurements without the ground based observation on Earth. However, in the abstract, the author does not express clearly and simply their purpose. Further, I suggest the authors focus on the Taiji mission since the there are other space-based gravitational wave detection mission with different formation with more spacecrafts larger than 3.
Lines 18-34: As a reader of your paper, I do not think she/he know the information about the Taiji formation or LISA formation. I think it is necessary to introduce the fundamental knowledge of the Taiji formation such that the reader has a clear picture about the Taiji formation.
Why the detection band of the Taiji mission is 0.1 MHz to 1 Hz?
In line 22, why the ground observations do not contain the deep space network?
In line 27-28, why the authors say that these methods cannot track the SC well?
In line 28-34, The authors say that each SC in the space-based GW detection formation is equipped with a high-precision laser inter-satellite link measurement sensor, what is the sensor, and what is the measurement the sensor can provide? Further, in the paper, the authors consider the other onboard measurements, such as the radial velocity measurement from spectrometer and the Sun-pointing angle measurement from Sun sensor, I suggest that the authors should introduce these onboard measurements. Finally, the motivation of the study is not present clearly at here.
Lines 35-68: The authors introduce the research status of the autonomous state estimation in line 35-51, then introduce that because of the orbital features of Taiji mission, the observability of the system using the inter-satellite measurements is poor since the difference in eccentricity or inclination of SCs orbits is small.
I suggest that in the two paragraphs, the author should introduce at first the corresponding research status of the orbit determination (or state estimation) of the Taiji/LISA formation, tell the reviewer the main difference of our study from the previous work. Based on the orbital features of Taiji mission, introduce the problem we face when we study the autonomous state estimation of Taiji borrowing from the experience of the current research status of the autonomous state estimation.
Lines 69-82: The authors say in line 70-72that “We focus on the Taiji project as a case study and specifically address state estimation in a multi body perturbation environment.”, I suggest the authors do not take the Taiji project as a case study, because in the title, the readers think the work of the paper is about Taiji, although the results of the work also apply to LISA, and others.
In line 72-74, “Considering the payload configuration and sensor costs, we establish two observation schemes to evaluate the necessity of the radial velocity sensor.” Please explain what are the two observation schemes.
In line 79-80, “Through our research, we achieve an estimated accuracy of approximately 3 km for the absolute position of the SC and approximately 0.1 m/s for the absolute velocity” Please the authors tell us the previous information about the state and the measurement accuracies of the observations.
Line 118: “r_rj=r_pj-r_i”, whether it is meaning that r_rj is the same for all i when j is fixed. If it is the same, I cannot understand it. If it is not the same, the authors should express the index i clearly in the left-hand side of the equality.
Lines 119-120: “w(t) represents the process noise of the system, and v(t) represents the measurement noise of the state” is the same as that given in lines 103-105, thus it is unnecessary to write it at here.
Lines 154-155: What are min individual and max individual? I suggest the smallest and largest.
Lines 158-160: “A decrease in the OI value or an increase in the CN value indicates a deterioration in the invertibility of the system’s observability matrix, implying a weaker observability of the system.” The language should be improved.
Lines 200-201: “The Table 5 below presents additional observability metrics for the system at different levels of measurement accuracy” I suggest the authors use OI and CN clearly.
Lines 248-249: In formula (30), why there are two values for the same i. And what are the values when i=nx+1, …, m.
Page 12-13: From the Figures 3-5, I suggest the authors do the statistics in Table 7 using the data when the estimation is stable, especially in figure 5.
Line 292-298: I suggest the authors analyze the results from the measurements’ viewpoint.
Lines 300-315: I suggest the authors correct the conclusion part according to the previous changes.
Please see the Comments and Suggestions for Authors.
Reviewer 2 Report
thank you for an interesting and well written paper.
1. I have not found designations of symbols (a, e, i, etc.) used in the heading of table 1. It is worth adding the discriptions.
In line 184 there is a misprint "the the"
Round 2
Reviewer 1 Report
I want to thank the authors for this accurate revision. All critics have been addressed and previous issues have been corrected/clarified. I recommend the paper for publication.